# DINOSTRAT version 2.1-GTS2020

Peter K. Bijl

Department of Earth Sciences, Faculty of Geosciences, Utrecht University, Princetonlaan 8A, 3584 CB Utrecht, the Netherlands

*Correspondence to*: p.k.bijl@uu.nl

**Abstract.** DINOSTRAT version 2.1-GTS2020 is now available (Bijl et al., 2024b); http://dx.doi.org/10.5281/zenodo.10506652). This version updates DINOSTRAT to the Geologic Time Scale 2020, and new publications are added into the database. The resulting database now contains over 9,450 entries from 209 sites. This update has not led to major and profound changes in the conclusions made previously. DINOSTRAT allows full presentation of first
and last stratigraphic occurrences of dinoflagellate cyst subfamilies and families, as well as the evolutionary turnover throughout geologic history, as a reliable representation of dinoflagellate evolution. Although the picture of dinoflagellate evolution from DINOSTRAT is broadly consistent with that in previous publications, the underlying data is with DINOSTRAT openly available, reproducible, and up to date. This release of DINOSTRAT allows calibration of stratigraphic records to the Geologic Time Scale 2020 using dinoflagellate cysts as biostratigraphic tool.

## 1 Introduction

DINOSTRAT version 1.1 (Bijl, 2022) offers a comprehensive open access database of organic-walled dinoflagellate cyst (dinocysts from hereon) first and last occurrences (hereafter referred to as 'events'). The dinocyst events in DINOSTRAT were calibrated to the Geologic Time Scale (GTS) 2012 (Gradstein et al., 2012). In 2020, a new geologic time scale was published (Gradstein et al., 2020). I here release DINOSTRAT version 2.1, in which stratigraphic ranges of dinocysts are calibrated to
the GTS2020 of (Gradstein et al., 2020).

## 2 Methods

In the release of version 2.1 of DINOSTRAT, I updated stratigraphic calibrations of dinocyst first and last occurrences to the GTS2020 (Gradstein et al., 2020). New papers that meet the criteria as laid out in DINOSTRAT version 1.1 (see Bijl, 2022 for more information about the approach) were added (see Table 1 for the new additions). The approach of data entry remained
the same in this new release. Published dinocyst ranges were included when they could be calibrated to a chrono-, bio-, or magnetostratigraphic zonation, whereby 5 reliability tiers were distinguished: 1) dinocyst events were calibrated against reliable magnetostratigraphy, 2) against compromised magnetostratigraphy, 3) against biostratigraphic zonations (nannofossils, foraminifer zones, ammonites), 4) indirectly against biostratigraphic zones (whereby the origin of those zones

is not exactly clear, or these zones were not derived from the same material), and 5) from unclear chronostratigraphic
constraints (Bijl, 2022). I had the paleolatitude evolution of the "stable" sites (i.e., away from major plate boundaries or teconically active regions) recalculated based on the upgraded paleolatitude.org (last access 12-1-2024) calculator, using the framework of Vaes et al. (2023). I upgraded the plotting tools with extra features: the stages from the GTS2020 (Gradstein et al., 2020) are added to visually aid the interpretation of the plots.

## 3 Results: DINOSTRAT version 2.1-GTS2020

The database is released as version 2.1-GTS2020 on Github and Zenodo (Bijl, 2024b); http://dx.doi.org/10.5281/zenodo.10506652), on which the new datasets and plotting tools can be found. The supplements (Bijl, 2024a) contain all the new figures for families, genera, species and sites in DINOSTRAT. The calibration of some entries in DINOSTRAT version 1.1 had to be adjusted for DINOSTRAT version 2.1-GTS2020, because of a revision of the Cretaceous ammonite zonations in GTS2020 (see Gradstein et al., 2020; Chapter Cretaceous). An overview of the changes induced by this
can be found on Github (Bijl, 2024b). This update has not led to any major changes in the database. For most events, this update caused only minor adjustments in the calculated absolute age. The adjusted ammonite zonation scheme in the Cretaceous that GTS2020 proposed did lead to a lot of recalibrations of dinocyst events. Yet, their absolute age should in principle not have changed much, since the ammonite zonation adjustment was mostly about redefining the index species and zone boundaries. With this update of DINOSTRAT, dinocyst events can now be used as chronostratigraphic indicator on the
GTS2020 time scale.

**Table 1. New papers added to DINOSTRAT version 2.1-GTS2020. Reference, Geography, Age base and Age top (in Ma), Tier (see DINOSTRAT version 1) and means of calibration to the Geologic Time Scale. For the meaning of the acronyms of the microplankton zones indicated in the column "Calibrated to" the reader is referred to the GTS2020**
**(Gradstein et al., 2020).**

| Reference | Geography | Age base | Age top | Tier | Calibrated to: |
|---|---|---|---|---|---|
| Bujak et al. (2022) | Arctic | 205 | 140 | 3 | Calibrated to boreal ammonite stratigraphy |
| Crouch et al. (2022) | Zealandia | 63 | 56 | 3 | Calibrated to NP nannoplankton stratigraphy |
| Gonzalez Estebenet et al. (2021) | Austral Basin | 70 | 54 | 3 | Calibrated to NP nannoplankton stratigraphy |

| | | | | | |
|---|---|---|---|---|---|
| Guerrero-Murcia and Helenes (2022) | Venezuela | 100 | 65 | 5 | Calibrated to stages |
| Jarvis et al. (2021) | Germany | 90 | 88 | 3 | Calibrated to boreal ammonite stratigraphy |
| Pearce et al. (2020) | UK | 100 | 90 | 3 | Calibrated to boreal ammonite stratigraphy |
| Pearce et al. (2022) | France | 86 | 70 | 3 | Calibrated to UC nannofossil stratigraphy |
| Thöle et al. (2023) | Antarctica | 0 | 0 | - | surface sediments |
| Torricelli et al. (2022) | Angola | 34 | 13 | 3 | Calibrated to CP nannofossil stratigraphy |
| Vasilyeva and Musatov (2022) | Crimea | 45 | 37 | 3 | Calibrated to CNE nannofossil stratigraphy |
| Vieira et al. (2020) | North Sea | 66 | 56 | 3 | Calibrated to stages with information from nannofossils |
| Vieira and Mahdi (2022) | Norway | 85 | 62 | 3 | Calibrated to stages with information from nannofossils |
| Copestake and Partington (2023) | North Sea | 202 | 140 | 3 | Calibrated to integrated microfossil and boreal ammonite stratigraphy |

## 4 Discussion

All the conclusions and interpretations made in the release paper of DINOSTRAT still stand with the addition of data and recalibration to the GTS2020. The North Atlantic Ocean is still strongly overrepresented in DINOSTRAT, and the Pacific
Ocean underrepresented in the geographical distribution of sites used to calibrate dinocyst assemblages (Fig. 1), and equatorial regions and the Southern Hemisphere remain underexplored, in spite these being large ocean regions (Fig. 2). Bujak et al. (2022) added crucial calibrated dinocyst events from the Jurassic Arctic Basin. The addition of the new Antarctic-proximal surface sediment data of Thöle et al. (2023) adds geographic distribution information for modern dinocysts in the Southern Hemisphere, but stratigraphic first occurrence of many modern species remains largely unknown (see Supplements).
Further exploration of DINOSTRAT has allowed the presentation of the stratigraphic ranges of dinocyst subfamilies and families (Fig. 3). As in Fig. 2b, Fig. 3 presents counts of oldest first occurrences and youngest last occurrences of dinocyst genera and species, but per subfamily or family, which yields the calibrated age of origination and final extinction of dinocyst

sub-families. This is a robust indication of the minimal stratigraphic range of these dincyst subfamilies. It must be noted however that the suprageneric classification will remain in a state of flux for the coming decades, as many uncertainties remain.

This may change the ranges of families and subfamilies described herein.

Peridiniphycideae have their first occurrence in the Ladinian (Middle Triassic), followed by Rhaetogonyaulacaceae, Suessiaceae and several miscellaneous Dinophyceae in the Carnian (Upper Triassic). The Heterocapsaceae, as first representatives of modern Peridiniales, the Mancodiniaceae as first representatives of the modern (but now non-cyst-forming) Cladopyxiineae, Nannoceratopsiaceae and Comparodiniaceae first occur in the Sinemurian (Lower Jurassic).

Scriniocassiaceae and Cladopyxiaceae (both part of the suborder of Cladopyxiineae) first occur in the Pliensbachian, which sees the last occurrence of the Shublikodiniaceae, the first of the cyst-forming dinocyst families to go extinct. The Toarcian sees the last occurrence of the Suessiales, as well as the first occurrence of the Pareodiniaceae and Leptodinioideae of the family Gonyaulacaceae, with many modern representatives. The Aalenian (Middle Jurassic) sees the first occurrence of the Cribroperidinioideae. The Gonyaulacoideae first occur in the Bajocian, during which also many other Gonyaulacaceae taxa

have first occurrences. The Scriniocassiceae have their last occurrence in this stage. The Stephanelytraceae and Areoligeraceae first occur in the Bathonian, while the Mancodiniaceae have their last occurrence. The Peridiniaceae first occur in the Callovian. The Deflandreoideae first occur in the Oxfordian (Upper Jurassic). The Nannoceratopsiaceae have their last occurrence and the Ceratiaceae their first occurrence in the Kimmeridgian. The Goniodomaceae have their first occurrence in the Tithonian. Palaeoperidinioideae first occur and Dollidiniaceae and Comparodiniaceae have their last occurrence in the

Valanginian (Lower Cretaceous). Stephanelytraceae have their last occurrence in the Aptian. Heterocapsaceae last occur in the Albian, however, motile cells still live in the modern ocean (Fensome et al., 1993). In the Cenomanian (Upper Cretaceous) Dinogymnioideae first occur, and Pareodiniaceae have their last occurrence. The Santonian sees the first occurrence of the Protoperidinioideae. The Ceratiaceae have their last occurrence at the end of the Maastrichtian. In the Danian (Paleocene), Wetzelielloideae have their first occurrence, the youngest subfamily to appear in the cyst record. Dinogymnioideae have their

last occurrence in the Selandian, however motile cells of this subfamily still occur in the modern ocean (Fensome et al., 1993). The Palaeoperidinioideae, Cladoxyxiaceae and Wetzelielloideae have their last occurrence in the Chattian (Oligocene). The Areoligeraceae have their last occurrence in the Burdigalian (Miocene). The Tortonian sees the last occurrence of the Leptodinioideae, and the Messinian the last occurrence of the Deflandreoideae.

The counts of oldest first occurrences and youngest last occurrences through time (Fig. 3) provide a rough indication of the

90 amount of turnover in subfamilies. Yet, for several reasons I dictate caution not to overinterpret Fig. 3, particularly the implications of cyst species turnover for biological diversity and evolution. The first practical reason is that many entries in DINOSTRAT have only a first occurrence or last occurrence logged, not both (because of their long range or because of calibration limitations), which prevents assessing the full stratigraphic range of many species, and therefore, prevents the assessment of total diversity through time. The second reason for this caution is the chance of absence bias or false negatives

in the fossil cyst record, for instance the absence of 4inocysts species because the corresponding dinoflagellate ceased to make cysts and not because it went extinct. Only about 10% of modern dinoflagellates include a cyst-phase in their life cycle (Bravo

and Figueroa, 2014). It is impossible to assess whether this fraction of cyst-producing dinoflagellates has changed through time. Yet, the representation of many dinoflagellate families in the fossil record of dinocysts (Fig. 3) does suggest that a cyst-phase was more common in a more diverse group of dinoflagellate taxa in the past. Interpreting what 5inocysts turnover would mean for the paleobiology of dinoflagellates remains complex due to the apparent ability for a few 5inocysts families to switch cyst formation on and off: the Cladopyxiaceae and Ceratiaceae are known as modern motile cells (Fensome et al., 1993) yet their stratigraphic occurrence as cysts in the fossil record ended long before modern times (Fig. 3). This demonstrates that dinocysts diversity likely underestimates past dinoflagellate diversity, by an unknown amount that is likely not stable through time. Finally, dinocyst taxonomy is a morphological taxonomy, and not defined on genetic grounds. On a cyst genus level and higher, the defining morphological differences focus on plate tabulation (Fensome et al., 1993), a feature so fundamental to cell structure that it probably does reflect biological diversity. The defining features of many cyst species, however, are focused on ornament rather than tabulation, and for some of those features it remains a question to what extent it reflects biological diversity, eco-phenotypical variability or effects of bio-provincialism. Examples of this complexity was a study whereby one strain of *Gonyaulax spinifera* produced cysts of *Spiniferites* and *Nematosphaeropsis* spp., with a full suite of morphological intermediates (Rochon et al., 2009). Such examples call into question the biological significance of cyst species, also in the fossil record. Although surely there is a signal of biological diversity captured in fossil dinocyst species assemblages, the total species diversity needs to be treated with some caution.

The stratigraphic ranges of dinocyst subfamilies or families (Fig. 3) from DINOSTRAT match closely those in the dinocyst diversity plot of (Fensome et al., 1996). This was expected, as both syntheses probably use overlapping literature resources as basis for the inferred dinoflagellate evolution. DINOSTRAT delivers a verifiable, reproducible database, that underpins dinoflagellate evolution, and that is updated to the most recent Geologic Time Scale. It provides a platform that allows iterative improvement of the communities' collective knowledge of dinocyst biostratigraphy. The similarity to other overviews of evolution (Fensome et al., 1996) demonstrates the completeness of the DINOSTRAT database.

## 5 Data and code availability

DINOSTRAT Version 2.1-GTS2020, including R-code to make the figures, is available on Github and Zenodo (Bijl, 2024b); http://dx.doi.org/10.5281/zenodo.10506652.

## 6 Supplements

Supplementary figures are available (Bijl, 2024a) for those not familiar with R programming:

- Paleolatitude distributions of dinocyst subfamilies and species (2066 plots)
- Paleolatitude distributions, stratigraphic ranges and distribution maps of dinocyst genera (471 plots)
- Paleolatitude distributions, stratigraphic ranges and distribution maps of dinocyst families (27 plots)

- Paleolatitude distributions of modern dinocyst species (96 plots)
- Stratigraphic ranges of dinocysts per site (211 plots)

## 7 Competing interests

The contact author has declared that none of the authors has any competing interests

## 8 Acknowledgements

The author thanks the *LPP* foundation and the department of Earth Sciences of Utrecht University for supporting this work. Thanks to Appy Sluijs, Henk Brinkhuis, Francesca Sangiorgi, Denise Kulhanek, Jeremy Young and Paul Bown for fruitful discussions. I thank the editor Giuseppe Manzella for helpful comments on this manuscript. Jan Hennissen, Rob Fensome and

an anonymous reviewer are thanked for their constructive review of the paper.

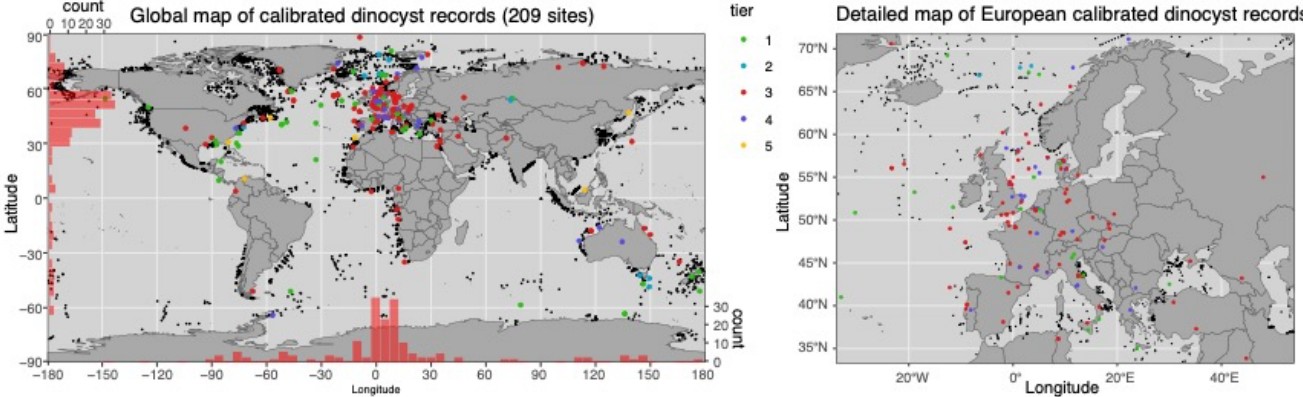

Figure 1: Map of sites included in DINOSTRAT Version 2.1-GTS2020. Maps produced using ggplot in R (Wickham, 2016).

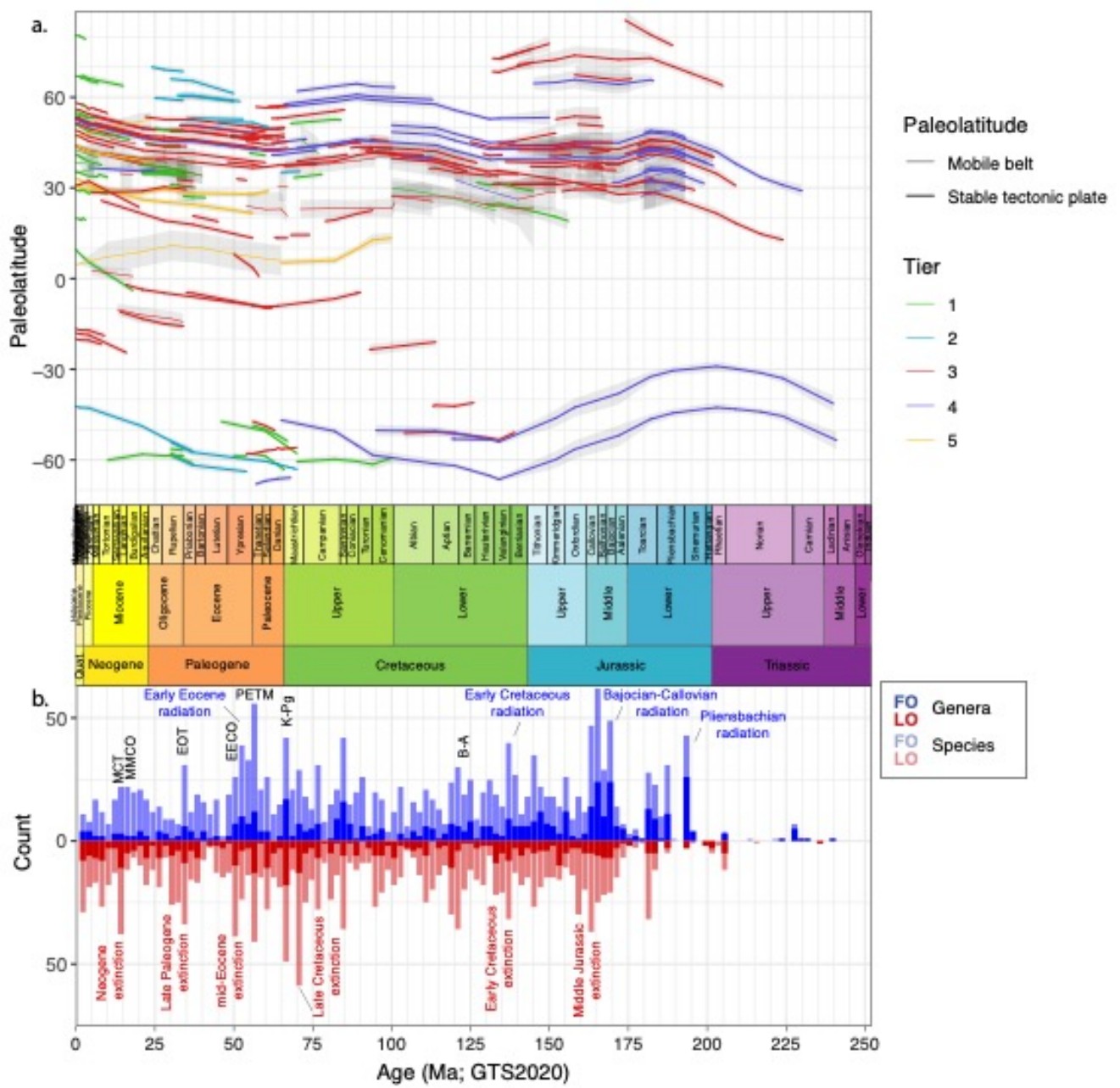

Figure 2: a Paleolatitude through time of sites used in DINOSTRAT Version 2.1-GTS2020. b. The number of oldest first occurrences and youngest last occurrences of dinocyst species, plotted in 2Myr bins. Plot produced using ggplot in R (Wickham, 2016).

# DINOSTRAT: Dinoflagellate cyst biostratigraphic ranges

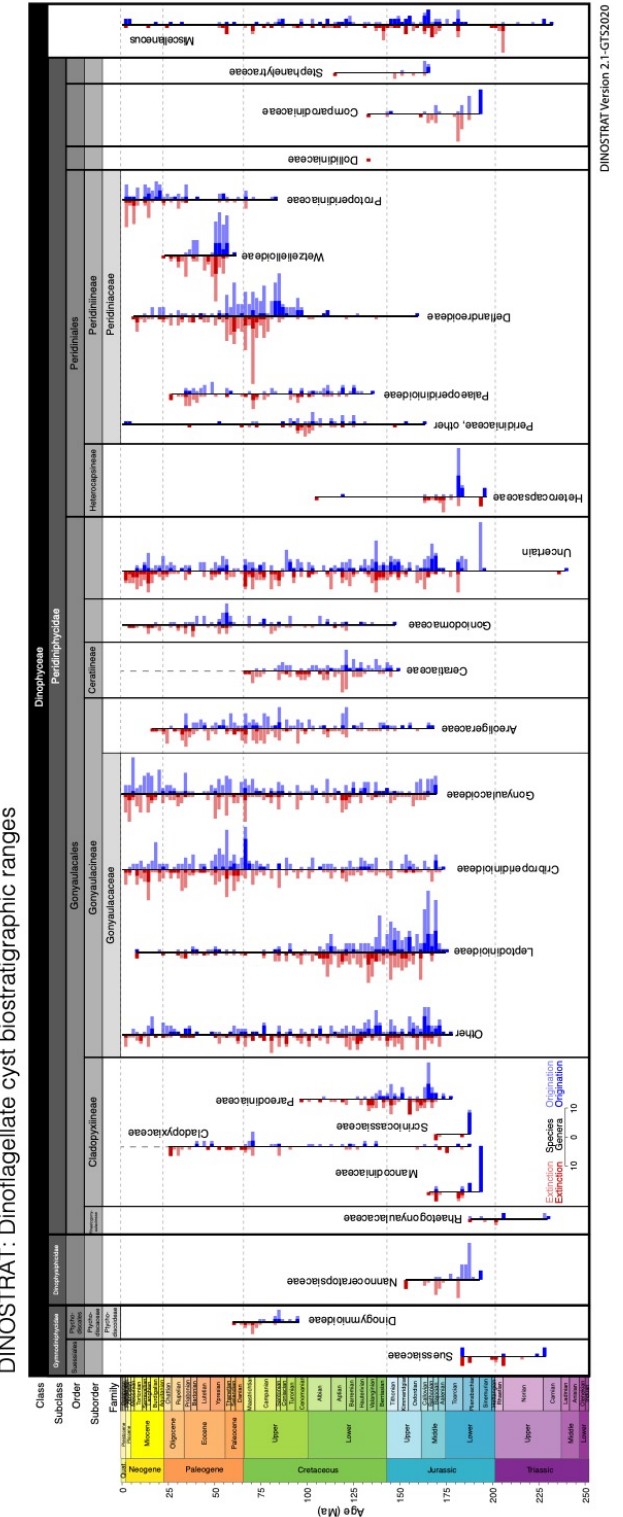

**← Figure 3: Number of dinocyst species and genera oldest first occurrence and youngest last occurrence, plotted per dinocyst subfamily or family, in 2 Myr bins.**

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
