# Peer review of "DINOSTRAT version 2.1-GTS2020"

_Earth System Science Data, 2023_

## Author Response (AR1)

**RC1 Anonymous**

In this short manuscript that accompanies the large dataset on the stratigraphic occurrences of dinoflagellate cysts, the author explains and summarizes how the (previously published) database has been updated in order to include newly-published data and, most significantly, to conform to the most recent version of the geological time scale, issued in 2020 (the GTS2020). Thus, the manuscript and the accompanying datasets (or, rather the other way around!) constitute an important contribution for future use by the larger dinocyst and biostratigraphy communities. The dataframes are easily accessible and accompanied with an explanatory "readme"-file and an R-script that illustrates how data from the dataset can be extracted and plotted. Being formatted as .csv-files, they are furthermore easily readable with other productivity software. While I do not have the expertise to assess the accuracy of the entire dataset itself, I can only recommend this update to be published as soon as possible, noting the one specific comment and a number of technical remarks given below.

Reply: I thank the reviewer for the positive response to the paper

**Specific comment.**

L68-69: concerning the "last occurrence of the Suessiales": I am aware of at least two cyst-forming dinoflagellate species belonging to the Suessiales that are extant today (found in sea ice samples, surface sediments and sediment traps): *Polarella glacialis* and *Biecheleria baltica*. I noted that *Biecheleria baltica* is placed within the Peridiniaceae, and *P. glacialis* under "misc" in the "modernsp.csv" dataframe, but this is incorrect. See, for instance, Montresor et al. (1999, Journal of Phycology 35, 186-197), Moestrup et al. (2009, Phycological Research 57, 203-220), or Limoges et al. (2020, Marine Micropaleontology), or Algaebase.

Reply: This suggestion is inconsistent with phylogeneitc relations hips from transcriptome work of Janouskovec et al., 2017, which is followed by Fensome et al., 2017, the suprageneric classification of which I here follow. No changes made.

**Technical remarks.**

L19: only the year of publication should be between brackets here (unless journal guidelines specify otherwise here?): … to the GTS2020 of Gradstein et al. (2020).

Same for L53, L55

L29: …see Bijl (2022).

Table 1: do the references need to be between brackets in the first column?

L50: try rephrasing "addition of additional data"? Perhaps "addition of new data"?

L53: would suggest rephrasing slightly: … equatorial regions and **the** Southern Hemisphere remain *(plural)* underexplored, in spite of **these being** large ocean regions.

(or something along those lines)

L55: *in* or *from* (not on?) the Southern Hemisphere

L56: again suggest slightly rephrasing: "… but the stratigraphic first occurrence still remains largely unknown for a large number of modern species."

L67: …Pliensbachian, which sees the last

(because Pliensbachian has become the subject of the sentence)

L87: caution must be **taken** (not taking). Do you mean to say "…, although caution must be taken here not to overinterpret the data." ?

L88: could you give some examples of such ''"previous overviews of dinoflagellate cyst evolution"?

L90: various sources give slightly different estimates for the % of cyst-forming dinoflagellates. I would suggest adding "about" or "ca." or something similar.

L93: not sure "scope" is the best word to use here. Also, "wider" than what? And "ubiquitous" would mean that cyst-formation was present everywhere. Perhaps this bit can be rephrased slightly (e.g., …common among a large number of… or a wide range of…)

L99: Clado**py**xiaceae (y missing after p)

In the README.md file, there is a typo in line 12: "sediemntary" (--» sedimentary). Also, line 36 mentions the "GTS2021", but this should be "GTS2020"? Finally, Mertens et al. (2014) is not in the list of references.

Reply: I followed all of these suggestions. I thank the reviewer for the thorough check of the data and the paper.

**RC2 Jan Hennissen**

This short manuscript provides an update to the DINOSTRAT database to ensure its compliance with the Geological Time Scale 2020 (Gradstein et al., 2020). This especially impacts the Cretaceous because of the introduction of an improved ammonite zone calibration. The manuscript is accompanied by databases detailing the palaeolatitudinal, geographic and stratigraphic distribution of dinoflagellate cysts through time. The Dinostrat database already is and will continue to be an invaluable tool for stratigraphic palynologists. The files are provided in .csv files which are easily imported in most software for statistical processing. The author provides a very useful script for the open-source R environment, capable of generating plots that allow the interpretation of stratigraphic ranges and geographical distribution of dinoflagellate families, genera, and species.

I recommend this manuscript is published subject to technical corrections below.

Author thanks Dr Hennissen for the positive review of the paper.

**General comments**

Checking the references of the underlying data reveals an exclusive reliance on academic sources. Recently, Copestake and Partington (2023) released a very detailed overview of Jurassic–earliest Cretaceous dinoflagellate cyst biozonation calibrated against the newest GTS and ammonite zonation. Their zonation also incorporates previously confidential, but now freely available industrial reports (through portals like https://ndr.nstauthority.co.uk/). Has the author considered incorporating their biozonation or is there no intent to look at reports from other than strictly academic sources?

Reply: DINOSTRAT indeed holds the peer-reviewed literature as basis for its content. In those resources, the tie to other age control is usually most clear. In the personal opinion of the author, most industry resources miss the verifyable tie to the international time scale. At the same time there is no reservation to include industry reports into the database, as long as these would pass through the decision tree. As for the paper of Copestake and Partington, 2023, the database has been updated up to Feb 2023 (the paper has been in initial submission stage for a long time), but in this revision, updates were implemented up to December 2023, and we included this good suggestion tot he database.

Changes made: Copestake and Partington, 2023 is added to DINOSTRAT as suggested.

**Minor comments**

Line 29:  change to "…chronostratigraphic constraints (Bijl, 2022)." Line 23 already mentions "for more information" and is therefore superfluous in Line 29.

Line 38–42: very long sentence. Could be split quite easily.

Table 1: check format of references. Change to format Bujak et al. (2022).

Line 53–54: Bujak et al. (2022)

Line 55: Thöle et al. (2022)

Line 55: …**in** the Southern Hemisphere

Line 62: din**oflagellate cyst sub-**families.

Line 82: insert comma after "however"

Line 82: "swim around". Consider changing the wording. I am not sure that the locomotion of a single celled organism using two flailing flagella qualifies as swimming.

Line 87: change to: …turnover within that subfamily, although caution must be taken to not overinterpret the data.

Line 88: insert comma after "evolution"

Line 89: Check the use of "dinocyst". It is preferable to consistently use "dinoflagellate cyst". If the author wishes to shorten this to "dinocyst" do so early on and make it uniform throughout. For example: dinoflagellate cyst (dinocyst from hereon).

Line 103: insert "the" in front of "genus">

Reply: All of the minor suggestions above were followed. Dinoflagellate cyst was changed to dinocyst throughout the paper, and the abbreviation properly introduced in the introduction. In the abstract, dinoflagellate cyst is not abbreviated.

**Comments on the R files**

The variable labels (column names) should have a uniform naming convention for ease of use (capitalized and no use of spaces as this can trigger errors in cross referencing). This is best done 'at source', namely in the csv files prior to loading in R. Some errors were triggered when trying to rename columns in loaded tibbles because of slight differences in spelling or syntax (see below).

Reply: For the revised manuscript, I made Branch Version-2.1-GTS2020, in which I uncapitalized all column names (except FO_LO) and changed this throughout the code and all csv files.

Packages jcolors and rgeos are archived R packages and cannot be retrieved from CRAN. It may be useful to provide a link in the code for manual download of the tar.gz files which users can then install.

Reply: Added in Version-2.1.

Because the script will produce files on the hard drives of the user (> 3000 files when the script finishes!!), it may be courteous to include a warning in the first line of the code. Below, the author could include a "setwd" prompt for the users. This will allow users to define a working directory where the output of the code will be saved. If this is not included, less experienced R users will discover in excess of 3000 pdfs in their root R folder.

Reply: Added warnings and a prompt to set directory in Version-2.1.

Line 99 (of the code): the author is trying to subset a column that does not exist ("RGB"). I assume this should be "CMYK"?

Reply: fixed this bug in Version-2.1.

Line 103 (of the code): I think this should read "right_join(modernst, clrs, by= "stage")" because the tibble "stg" has not been defined.

Reply: fixed this bug in Version-2.1.

Line 124 (of the code): the link to the 2023 database does not work (yet?). I manually loaded the csv the author provided after which the figures were plotted.

Reply: It works now in Version-2.1

"modernst" is defined in line 41 (of the code), but also in line 104 (of the code). These two instances refer to two different datasets. The former to https://raw.githubusercontent.com/bijlpeter83/DINOSTRAT/main/data/modernst.csv, the latter to https://raw.githubusercontent.com/bijlpeter83/DINOSTRAT/main/data/stages.csv. This causes problems on line 295 (of the code) when "modernst" is called but it has already been overwritten and the run terminates.

Reply: It works now in Version-2.1

Many of the plots involving grouping in ggplot's geom_line () generate a long list of warnings because the "group" aesthetic is used for a single observation. This warning can also be triggered by using a factor variable in the X axis. These warnings have no impact on the final plots, but if possible, check if the "group" aesthetic is strictly required (given the groups consist of a single observation) if not, try setting "group = 1" to remediate.

Reply: some groups do consist of one observation, some do not, which is why I do not think this can be fixed. As these warnings do not really impact the output, I left it as is.

**References**

Copestake, P., Partington, M.A., 2023. Chapter 13. Biozonation of the Jurassic–lowermost Cretaceous of the North Sea region. Geological Society, London, Memoirs 59, M59-2022-2061 doi:10.1144/M59-2022-61.

Gradstein, F.M., et al., 2020. Geologic Time Scale 2020. Elsevier

**RC3 Rob Fensome**

Dear editor

I have read the manuscript by Peter Bijl. It is a description of a database that is potentially very useful and I applaud any initiatives to improve the utility and consistent application of our fossils. As such the manuscript constitutes an important contribution. However I think it should undergo moderate changes before acceptance. I have tried to help smooth out the writing in places, which I hope helps. I have also annotated numerous comments that I hope will be useful. Most seriously, some misstatements need to be corrected.

I have the following supplementary comments that are generally, if not strictly, in manuscript order.

   1)  I recommend the use of the active voice wherever possible. It makes the text clearer.

Reply: I adjusted this where I thought this would help

2) I recommend spelling out "families and subfamilies" rather than the possibly ambiguous and awkward "(sub)families.

Reply: I changed this

2) I think the real value of such a compilation is at the species level and possibly at the family level. The classification and I think subfamilies are more in a state of flux and morphological limits (and thus ages) are likely to be modified in the future.

Reply: I agree with the reviewer that not on all taxonomic levels the combination of data is equally meaningful, and that taxonomic grouping will remain in a state of flux for some time. This is exactly why I made the database flexible enough to be easily adjusted to those changes. I added the cautionary note to the paper, that on genus, subfamily and family level the taxonomic groupings will remain in a state of flux.

3) The name Shublikodiniaceae has long been rejected in favour of Rhaetogonyaulacaceae, which is formally conserved ... see https://onlinelibrary.wiley.com/doi/abs/10.2307/1223595

Reply: I changed this in Version 2.1 of the database and in Figure 3

I need convincing that areologeraceans occur as early as the Toarcian. This is where the data needs more critical scrutiny (species identities and assignments, etc.).

Reply: I checked this in the database and noted that this was an error in the text. In the database, the oldest Areoligeraceae occur in the Bathonian. Figure 3 shows the FO correctly.

I don't now think Mesozoic "heterocapsaceans" are related to modern *Heterocapsa,* which is late derived. I think we mentioned this in Janouskevic et al. in 2017.

Reply: I changed this in Figure 3.

By "previous overviews" on line 140 you would obviously include MacRae et al 1996. If so, you obviously haven't read our paper. Although admittedly I haven't checked, I'm sure we had the same provisos, which are truisms ... so the authors should be careful in claiming exceptionalism here. My view has always been to try to find the patterns expressed by the fossil record and investigate what they might demonstrate, notwithstanding all the provisos that the authors are repeating from earlier works.

Reply: I rephrased. I understand that the sentence could be read to mean something I wasn't intending to say.

Re lines 152 ff, Nannos and Dinogyms … do not have modern representatives. The clados are an iceberg group showing a decline in diversity matching relatively few species today. The best example of what you are trying to express is that of the Ceratiaceae. Another point is that dinogymniaceans may not have been (conventional) cysts (see Fensome et al. 1993).

Reply: I changed this sentence to focus on this phenomenon in Cladopyxiaceae and Ceratiaceae.

Re lines 154 ff, I strongly disagree that the morphological range that we see in the fossil record is completely abiological or random. Patterns of morphological distribution are critical, and are our primary way of trying to understand past phylogeny. I think the authors are conflating the idea that individual species concepts are somewhat arbitrary with the idea that morphological variation says nothing about evolution. Fossil dinoflagellate

morphospace has huge mountains and valleys that, in my view, clearly reflect aspects of evolution.

I realize that some of the following sentences seem to address this criticism, but the writing in my view could be clearer and less convoluted on these aspects.

Reply: I never intended to convey that all dinocyst morphology is abiological. The text also does not mention this at all, but the insinuation of the reviewer is sufficient to reconsider the text, as other readers might come to the same impression. The abundance of evidence for eco-phenotypical variability in dinoflagellate cysts produced by the same dinoflagellate should in my view, and that of many others in our community, lead to a healthy bit of skepticism regarding the biologic significance of some fossil species. Although my view can be backed up with just as much evidence as the view of the reviewer, all I can do in the paper is to lay out this dispute objectively.

Please let the author know the source of these comments; on principle I generally do not agree to do unsigned reviews. And I would be happy to clarify or help if the author wishes.

Sincerely, Rob Fensome